# Ketogenic diet improves disease activity and cardiovascular risk in psoriatic arthritis: A proof of concept study

**Roberta Ramonda**[1ᵒ*], **Francesca Ometto**[2ᵒ*], **Giovanni Striani**[1ᵒ*], **Giacomo Cozzi**[1], **Daniela Basso**[3], **Filippo Evangelista**[1], **Mariagrazia Lorenzin**[1], **Laura Scagnellato**[1], **Ada Aita**[3], **Marta Favero**[1,4], **Filippo Brocadello**[5], **Andrea Doria**[1]

**1** Rheumatology Unit, Department of Medicine (DIMED), Padova University Hospital, Padova, Italy, **2** Rheumatology Outpatient Clinic, Local Health Unit 6 Euganea (Azienda ULSS 6 Euganea), Padova, Italy, **3** Laboratory Medicine, Department of Medicine (DIMED), Padova University Hospital, Padova, Italy, **4** Internal Medicine 1, Department of Medicine (DIMED), AULSS2 Marca Trevigiana, Ca' Foncello Hospital, Piazzale dell'Ospedale, Treviso, Italy, **5** Affidea Morgagni Polyclinic, Padova, Italy

ᵒ Statement of equal contribution: Roberta Ramonda, Francesca Ometto and Giovanni Striani contributed equally to this study.
* roberta.ramonda@unipd.it

## Abstract

### Objectives

Very low-calorie ketogenic diet (VLCKD) is a low-carbohydrate, low-calorie regimen that leads to rapid weight loss and may reduce inflammation. This study assessed the impact of VLCKD on anthropometric measurements, inflammatory biomarkers, metabolic health, and cardiovascular risk in psoriatic arthritis (PsA) patients moderately overweight or in class I obesity.

### Methods

A proof-of-concept single-arm monocentric study involved PsA patients undergoing a 9-week VLCKD treatment. Patients with Body Mass Index (BMI) ≥27 and <35, in stable (≥6 months) remission or low disease activity, as defined by Disease Activity in PSoriatic Arthritis (DAPSA) score, were included and underwent nutritional evaluations every 3 weeks. The study analyzed changes after the VLCKD intervention and the association between changes of anthropometric parameters and clinical and laboratory variables.

### Results

Twenty patients were enrolled since April 2022 and completed the study in May 2023. Median baseline BMI was 30.9 (interquartile range 29.1–33) kg/m². All participants exhibited low baseline disease activity, which correlated with BMI (Spearman's correlation coefficient ($r_s$)=0.59,p=0.007). Following VLCKD, significant improvements were observed in all anthropometric measures (BMI -3.5[-4;-2.6]), PsA activity (DAPSA -6.1[-16.8;3.7]), cardiovascular parameters (SCORE2 index -0.2[-0.7;0.1]), insulin resistance (Homeostatic Model Assessment-Insuline Resistance -2.1[-1.1;-3.0]), and lipid profile. Most inflammatory

**Data availability statement:** All relevant data are within the manuscript and its Supporting Information files.

**Funding:** The author(s) received no specific funding for this work.

**Competing interests:** The authors have declared that no competing interests exist.

biomarkers remained within normal limits. BMI reduction correlated with changes in DAPSA scores ($r_s$=0.52,p=0.020). Patients with higher baseline weight or clinical activity experienced more pronounced improvements.

## Conclusions

VLCKD significantly improved PsA activity and metabolic health. Patients with a higher BMI and less controlled disease are particularly motivated and could benefit more from VLCKD compared to those with lower BMI or better disease control.

## Introduction

Diet may influence both the onset and management of rheumatic diseases, contributing to the complex interplay between genetic and environmental factors. Specific nutrients can impact the balance between anti-inflammatory and pro-inflammatory cytokines: omega-3 polyunsaturated fatty acids (PUFAs) are known for their anti-inflammatory properties [1,2,3,4] whereas sugars and high-calorie foods may increase risk of rheumatic diseases [5]. Although few studies have investigated the influence of specific diets in rheumatic conditions, there is growing evidence of the importance of diet in patients with psoriatic arthritis (PsA) and spondyloarthritis, particularly the Mediterranean diet [6–10]. Furthermore, PsA patients have an elevated risk of overweight and should be routinely monitored for metabolic and cardiovascular disease [11–13,14]. A healthy and balanced diet is particularly recommended in rheumatic patients to achieve and maintain normal weight, as well as mitigate biomechanical joint stress [15,16].

The ketogenic diet (KD) is a low-carbohydrate, high-fat regimen that induces ketosis in the body and has become widely recognized as an effective tool for weight loss [17]. The very-low-calorie KD (VLCKD) is an extremely restrictive regimen providing very few grams of carbohydrates, low calorie intake and a minimal protein intake necessary to preserve lean mass [18]. The VLCKD and KD have been investigated for various clinical conditions beyond their established use in severe obesity and drug-resistant epilepsy [19,20], including cardiovascular disease [21], osteoarthritis [22], fibromyalgia [23], psoriasis [24,25], rheumatoid arthritis [26], and gout [27].

A series of intricate processes triggered by the KD suggests that this dietary approach could be an effective tool in reducing both inflammation and pain [20,28]. Beta hydroxybutyrate (BHB), the main ketone body, may exert inhibitory effects on various pro-inflammatory cytokines such as interleukins (IL)-1β, IL-17, IL-18. These cytokines have been shown to decrease with KD in various inflammatory conditions, including rheumatoid arthritis [19,29,30]. The KD and VLCKD also improve insulin resistance, which in turn inhibits the production of IL-1α, IL-1β, IL-6, tumor necrosis factor α (TNF-α), and leptin [31,32]. These mechanisms, besides downregulating pro-inflammatory cytokines produced by the adipose tissue, suggest that the anti-inflammatory effect of KD and VLCKD may be superior to that observed with other hypocaloric diets [33,34].

Therefore, we aimed to conduct a single-arm, proof-of-concept monocentric study to evaluate the effectiveness of a 9-week VLCKD on disease activity measured with Disease Activity in PSoriatic Arthritis (DAPSA) index in patients with PsA and moderate overweight status or class I obesity. Secondary endpoints were to measure the efficacy of VLCKD on anthropometric measurements, other measures of PsA activity, inflammatory indices, lipid profile, insulin resistance, and cardiovascular risk.

## Methods

### Study design

We conducted a single-arm, prospective, single-center proof-of-concept study at the Spondyloarthritis outpatient clinic of Padova University Hospital, Italy. Moderately overweight or class I obesity patients with PsA who underwent a 9-week VLCKD regimen were enrolled. Patients were evaluated every 3 weeks by a dietitian, who provided guidance to patients, and by a rheumatologist at baseline (W0) and at the end of the study (W9).

### Patients

Inclusion criteria were: (I) diagnosis of PsA according to the Classification Criteria for Psoriatic Arthritis (CASPAR) [35] with disease duration <10 years; (II) age ≥18 years; (III) body mass index (BMI) ≥27 and <35; (IV) stable treatment with conventional synthetic disease-modifying anti-rheumatic drugs (DMARDs) or biological/targeted DMARDs in the 6 months prior to enrollment; (V) remission or stable low disease activity (LDA) as defined by Disease Activity in Psoriatic Arthritis (DAPSA) index in the 6 months prior to enrollment. Exclusion criteria were: (I) concomitant musculoskeletal diseases; (II) patients unable to follow nutritional indications; (III) ongoing corticosteroid treatment; (IV) patients on a specific diet or regularly taking nutraceuticals or nutritional supplements; (V) contraindications to VLCKD [36].

### Diet protocol

Patients were assessed every three weeks by a nutritionist who inquired about their diet, provided nutritional guidance, and offered alternatives in cases of difficulty adhering to the nutritional protocol. A very low carbohydrate content (<20 g/day) was allowed. Protein and lipid intake were approximately 1–1.4 g/kg of ideal body weight/day and 15–30 g/day, respectively. Total caloric intake was between 450 and 800 kcal/day based on calculated ideal body weight.

### Collected data

A complete list of variables included in the study may be found in S1 Table. Sociodemographic data were collected at W0. At W0 and W9 the main variables collected were:

- *Anthropometric measurements:* body weight (to the nearest 0.1 kg), standing height (to the nearest 0.5 cm); abdominal circumference; BMI was calculated accordingly.

- *Measures of PsA activity (including patient-reported outcomes):* tender joint count (TJC) and swollen joint count (SJC), dactylitis; DAPSA, Disease Activity Score-28 – C Reactive Protein (DAS28-CRP), ASDAS-CRP (Ankylosing Spondylitis Disease Activity Score – C Reactive Protein), Psoriasis Area Severity Index (PASI), Spondyloarthritis Research Consortium of Canada score (SPARCC); physician- and patient-reported outcomes: lost work hours according to the Work Productivity and Activity Impairment Questionnaire (WPAI), VAS (Visual Analogue Scale) pain; patient and physician global assessment measured on a VAS scale (PtGA and PGA).

- Blood and urine samples: Samples were collected after an overnight fast and analyzed at the Laboratory Medicine facility of Padova University Hospital for inflammatory indices (hsCRP [high-sensitivity C-reactive protein], Erythrocyte Sedimentation Rate [ESR], IL-1α, IL-1β, IL-6, TNF-α, fecal calprotectin [f-CPT]), and for hematology and biochemistry parameters (lipid profile, insulinemia, hepatic, renal, and thyroid function, blood glucose, azotemia, protein electrophoresis, uricemia, intestinal permeability test; urinary ketones [as

an indicator of adherence to the VLCKD]). The Homeostatic Model Assessment for Insulin Resistance (HOMA-IR) index was calculated using the formula: fasting glucose (mg/dL) × fasting insulin (mU/L)/ 405. Due to reduced levels of IL-1α, IL-1β, IL-6, and f-CPT, these variables were converted into binary categorical variables for statistical analysis, using the upper limit of the normal range specified by the laboratory test company as the cut-off. The cut-offs were: 3.9 ng/L for IL-1α, 5.0 ng/L for IL-1β, 7.0 ng/L for IL-6, and 70 μg/g for f-CPT.

- *Nutritional questionnaires*: PREDIMED (PREvención con DIeta MEDiterránea, Mediterranean diet adherence questionnaire) and FFQ (Food Frequency Questionnaire).

- *Cardiovascular parameters:* SBP (Systolic Blood Pressure), DBP (Diastolic Blood Pressure), SCORE2 index (Systematic COronary Risk Evaluation 2 index). The SCORE2 index [37] was calculated using the appropriate calculators for individuals aged over 75 and those with type 2 diabetes. Subjects under 40 years of age were excluded from SCORE2 as per definition. Subsequently, the scores were adjusted by multiplying by 1.5, as recommended by the EULAR cardiovascular risk guidelines for arthritis [14].

Medical records were retrieved by a single investigator and recorded anonymously in an electronic database on a password-protected computer. The data were last accessed on January 8, 2024. All enrolled patients provided written informed consent to participate in the study in accordance with the principles of the Declaration of Helsinki, and for the anonymous use of personal data, in compliance with Italian Legislative Decree 196/2003. Ethical approval for the study was obtained on December 12, 2011 (Approval No. 2438P).

## Statistical analysis

The study analysis was structured into four phases: (I) association between variables at W0 that could impact the analysis of the effect of VLCKD; (II) changes in parameters between W0 and W9 to identify significant modifications following VLCKD; (III) association between changes in anthropometric parameters and other variables during the study to assess the impact of weight, BMI and abdominal circumference modifications on PsA disease activity; (IV) association of variables at W0 with modifications during the study to identify baseline predictors of a better response to the VLCKD.

Continuous variables were tested for normality using the Shapiro-Wilk test, revealing a non-normal distribution for all variables, necessitating non-parametric tests. The analysis of the changes in continuous variables between W0 and W9 was conducted using the Wilcoxon test. Associations between categorical variables were evaluated using either the Chi-square test or Fisher's exact test, as appropriate. The Kruskal-Wallis test was employed to assess the association between categorical and continuous variables. Spearman's correlation coefficient was utilized for continuous variables. All tests were considered statistically significant at the alpha level of $p < 0.05$. Continuous variables are expressed as median and interquartile range (IQR). SPSS Version 25.0 was used for statistical analysis.

## Results

### Characteristics of patients at W0

Twenty patients were included starting from April 27, 2022, with the last patient completing the study on May 31, 2023. None of the patients dropped out of the study or reported adverse events. The characteristics of the patients and assessments at W0 are reported in Table 1 and 2 and in S2–S5 Tables. The patients were 9 (45.0%) male with a median age of 55.5 (50–59.3)

**Table 1. Characteristics of the patients at W0.**

| Subjects | 20 |
|---|---|
| Male, n (%) | 9 (45.0) |
| Age, years, median (IQR) | 55.5 (50;59.3) |
| Height, cm, median (IQR) | 173 (164;177.3) |
| Smoke ever, n (%) | 6 (30.0) |
| Cardiovascular comorbidity, n (%) | 7 (35.0) |
| Type 2 diabetes, n (%) | 2 (10.0) |
| Disease duration, years, median (IQR) | 8.8 (4.2;14.3) |
| Current csDMARD, n (%) | 6 (30.0) |
| Current b/tsDMARD, n (%) | 14 (70.0) |

Categorical variables are reported as number and percentage, continuous variables are reported as median and inter-quartile range.

IQR, interquartile range; csDMARDs conventional synthetic disease modifying antirheumatic drugs; b/tsDMARDs biological/targeted synthetic disease-modifying antirheumatic drugs.

years, BMI 30.9 (29.1–33) kg/m$^2$. Seven patients (70%) presented with cardiovascular comorbidity and 2 (6.6%) had type-2 diabetes. Thirteen patients (65%) were treated with biological DMARDs. At W0, all patients were in PsA remission or low disease activity according to DAPSA: 11.5 (5.2;23.8). ASDAS-CRP was 1.5 (0,5;1,7) and PASI 0 (0;1.8). All inflammatory biomarkers tested in the study revealed no detectable inflammation at W0, except for 4 (21.1%) patients who had slightly elevated IL-6 levels (Table 2). Cardiovascular risk score was 6.9 (3.8;14), according to SCORE2 (expressed as 10-year risk of cardiovascular events).

## Association between variables at W0

The analysis of the association between measurements at W0 was conducted using different types of tests and is comprehensively detailed in S3–S5 Tables. At W0, the anthropometric measurements (weight, BMI, abdominal circumference) were significantly correlated with each other and male subjects had a significantly higher weight. BMI positively correlated with DAPSA and DAS28-CRP, indicating that greater disease severity at W0 was present in subjects with a higher BMI ($r_s$=0.59, p=0.007 and $r_s$=0.47, p=0.035, respectively). No significant associations were observed between the anthropometric measurements and inflammatory biomarkers. Measures of peripheral and axial PsA, and patient-reported measures, correlated with each other, except for SPARCC (S4 Table). As expected, DAS28-CRP correlated with ESR and hsCRP, while peripheral joint-related indices (SJC) and PGA were significantly associated only with IL-6 levels, which were elevated in those with a more active disease (S3 Table). Conversely, at W0, TNFα levels correlated negatively with joint disease indices, DAPSA and DAS28-CRP, with higher levels in those with less active disease ($r_s$=0.48, p=0.034 and $r_s$=0.45, p=0.047, respectively).. As, expected, fasting insulinemia, glucose and HOMA-IR positively correlated with weight, BMI and abdominal circumference, being higher in individuals with increased measurements (S4 Table). At W0, anthropometric measurements no significant associations with the lipid profile or the cardiovascular indices (S4 Table).

## Changes between W0 and W9 (identification of significant modifications following VLCKD)

The modification of the variables throughout the study is reported in Table 2 and is fully provided in S6–S11 Tables and S1 Fig. All anthropometric variables were significantly reduced

**Table 2. Characteristics of the patients and modification during the study.**

| | W0 | W9 | Δ (W9-W0) | p* |
|---|---|---|---|---|
| Weight, kg, median (IQR) | 91 (81.8;99.7) | 83.6 (70,5;88,5) | -10.2 (-12.6;-7.7) | <0.001 |
| Height, cm, median (IQR) | 173 (164;177.3) | 173 (164.8;177.3) | 0 (0;0) | NA |
| BMI, kg/m2, median (IQR) | 30.9 (29.1;33) | 27.2 (25.8;29.7) | -3.5 (-4;-2.6) | <0.001 |
| Abdominal circumference, cm, median (IQR) | 106 (103.8;115.3) | 96 (91;106) | -11.8 (-14;-10) | <0.001 |
| Tender joints count (0–68), median (IQR) | 3 (1;6) | 0 (0;2.5) | -1 (-2;0) | 0.007 |
| Swollen joints count (0–68), median (IQR) | 0 (0;2) | 0 (0;0.3) | 0 (-1.3;0) | 0.033 |
| DAPSA, median (IQR) | 11.5 (5.2;23.8) | 8.6 (2.1;13.5) | -6.1 (-16.8;3.7) | 0.006 |
| DAS28-CRP, median (IQR) | 2.6 (2.1;3.7) | 2.4 (1.8;2.8) | -0.9 (-1.8;0.7) | 0.068 |
| ASDAS-CRP, median (IQR) | 1.5 (0,5;1,7) | 0.9 (0.4;1.4) | -0.4 (-0.7;0) | 0.016 |
| SPARCC, median (IQR) | 1.5 (0;4.5) | 0 (0;4.3) | 0 (-3.5;2.5) | 0.072 |
| PASI, median (IQR) | 0 (0;1.8) | 0 (0;0.2) | 0 (-1.2;0) | 0.027 |
| Patient global assessment (0–10 cm), median (IQR) | 4 (2;5.3) | 3 (1;4.3) | -0.5 (-4;1.5) | 0.103 |
| Physician Global Assessment (0–10 cm), median (IQR) | 3.5 (2;5) | 2.5 (0.8;4) | -0.5 (-4;1.3) | 0.038 |
| WPAI – Lost work hours, median (IQR) | 0 (0;0) | 0 (0;0) | 0 (0;0) | 0.196 |
| hsCRP, mg/L, median (IQR) | 0.2 (0.1;0.4) | 0.2 (0.1;0.7) | 0 (-0.1;0.1) | 0.777 |
| ESR, mm/h, median (IQR) | 14 (7.8;30.8) | 19 (8.8;42.5) | 4 (-0.3;12) | 0.055 |
| IL-1α ≥ 3,9 ng/L°, n (%) | 0 (0) | 0 (0) | 0 (0) | NA |
| IL-1β ≥ 5,0 ng/L°, n (%) | 4 (21.1) | 1 (5.3) | -3 (-15.8) | 0.582 |
| IL-6≥7,0 ng/L°, n (%) | 4 (21.1) | 4 (21.1) | 0 (0.0) | 0.110 |
| TNFα, ng/L°, median (IQR) | 12.8 (7.3;92.9) | 12.5 (6.6;63.5) | -1.7 (-14.4;0.4) | 0.227 |
| Fecal calprotectin ≥70 µg/g^, n (%) | 7 (38.9) | 4 (27.8) | -3 (15.8) | 0.001 |
| Total cholesterol, mg/dl, median (IQR) | 194 (165;233) | 185 (135.3;209.5) | -23 (-41.5;-4) | 0.007 |
| HDL cholesterol, mg/dl, median (IQR) | 51.5 (46.3;62.5) | 52.5 (42.3;64.5) | -3.5 (-9.3;2.3) | 0.150 |
| LDL cholesterol, mg/dl, median (IQR) | 114 (96.5;152.3) | 110 (68.3;136) | -10 (-31.3;-2.5) | 0.037 |
| Triglyceride, mg/dl, median (IQR) | 103.5 (81.5;128.5) | 79 (65;89.5) | -25 (-50.3;-3.8) | 0.004 |
| Uricemia, mmol/L, median (IQR) | 0.4 (0.3;0.4) | 0.4 (0.3;0.4) | 0 (0;0) | 0.861 |
| Blood glucose, mg/dl, median (IQR) | 100 (87.5;115.5) | 95 (89.5;102.8) | -7 (-16.3;2.5) | 0.076 |
| Insulinemia, mU/L°, median (IQR) | 15.4 (11;21.8) | 9.2 (6.6;63.5) | -5.6 (-9.7;-3.9) | <0.001 |
| HOMA-IR, median (IQR) | 3.7 (2.4;5.8) | 2.3 (1.4;3.4) | -2.1(-1.1;-3.0) | <0.001 |
| Lactulose/mannitol ratio, median (IQR) | 0 (0;0) | 0 (0;0) | 0 (0;0) | 0.035 |
| Ketones, g/L, median (IQR) | 0 (0;0) | 0 (0;0.1) | 0 (0;0.1) | 0.035 |
| PREDIMED score, median (IQR) | 7 (7;9) | 8.5 (6.8;10) | 0.5 (-2.3;2) | 0.761 |
| Systolic blood pressure, mmHg, median (IQR) | 140 (130;145) | 130 (120;140) | 0 (-6.3;0) | 0.246 |
| Diastolic blood pressure, mmHg, median (IQR) | 85 (80;90) | 80 (80;90) | 0 (0;5) | 0.674 |
| SCORE2, 10-year risk percentage°§, median (IQR) | 6.9 (3.8;14) | 7.4 (4.1;12.3) | -0.2 (-0.7;0.1) | 0.009 |

°Data calculated from 19 patients; ^ data calculated from 18 patients.

IQR, interquartile range; BMI, body mass index; DAPSA, disease activity in psoriatic arthritis; DAS28-CRP, disease activity score on 28 joints with C-reactive protein; ASDAS-CRP, Ankylosing Spondylitis Disease Activity Score – C-Reactive Protein; SPARCC, Spondylarthritis Research Consortium of Canada; PASI, Psoriasis Area Severity Index; WPAI, Work Productivity and Activity Impairment questionnaire; hsCRP, High Sensitivity CRP, ESR, Erythrocyte Sedimentation Rate; IL, interleukin, TNFα, Tumor Necrosis Factor alpha; HDL, High Density Lipoprotein; LDL, Low Density Lipoprotein; HOMA-IR Homeostatic model assessment for insulin resistance; PREDIMED, PREvención con DIeta MEDiterránea; SCORE2 (Systematic Coronary Risk Evaluation 2).

with a median weight reduction of -10.2 (-12.6;-7.7) kg, a BMI reduction of -3.5 (-4;-2.6), and an abdominal circumference reduction of -11.8 (-14;-10) cm. Also, most domains of PsA (peripheral, axial, skin disease and patient-reported outcomes), significantly improved except for enthesitis (SPARCC). In particular, the DAPSA was reduced by a median of -6.1 (-16.8;3.7).

Inflammatory biomarkers remained stable following VLCKD, except for f-CPT levels which decreased significantly (Table 2). Among the laboratory variables, insulin levels significantly decreased by -5.6 (-9.7;-3.9) mU/L and accordingly HOMA-IR (-2.[-1.1;-3.0]). The lipid profile overall improved: total cholesterol (TC), Low Density Lipoprotein (LDL)-cholesterol, and triglycerides significantly decreased [by -23 (-41.5;-4), -10 (-31.3;-2.5) and -25 (-50.3;-3.8) mg/dL, respectively] with non-significant reduction in High Density Lipoprotein (HDL). As anticipated in the context of a ketogenic diet, urinary ketones increased significantly (Table 2). Lactulose/mannitol ratio increased due to a non-significant decrease in mannitol while lactulose and sucrose remained stable (Table 2, S9 Table and S1 Fig).

Nutritional questionnaires revealed a significant reduction in the consumption of cereal derivatives and processed cereal products, fresh fruit, legumes, dairy products, sweets, sodas, and alcoholic beverages. Conversely, the consumption of nuts, seafood, and eggs significantly increased (Fig 1). Adherence to the Mediterranean diet, measured by PREDIMED, remained stable (Table 2).

*Association between changes in anthropometric measurements and changes in other variables during the study (assessing the impact of weight, BMI and abdominal circumference modification on PsA disease activity)*

The results of the associations between the modification of anthropometric measures and modification of inflammatory and disease indices are reported in Fig 2. For full details, see S12–S18 Tables.

The reduction in BMI significantly correlated with a reduction of PsA activity measures, specifically peripheral arthritis (DAPSA, $r_s$ =0.52, p=0.020), and WPAI lost work hours ($r_s$ =0.26, p=0.022). The reduction in abdominal circumference correlated with the reduction of TJC ($r_s$ =0.48, p=0.033). No significant correlation between changes in anthropometric measures and inflammation indices was observed, except for a modest and inverse association between the change in abdominal circumference and the change in hsCRP ($r_s$=0.50, p=0.046) (S14 and S18 Tables). Similarly, the improvement in anthropometric measures was not associated with a consistent change in the lipid profile, while a reduction of weight and BMI correlated with reduction of blood glucose ($r_s$=0.60, p=0.005 and $r_s$ =0.62, p=0.003, respectively) and HOMA-IR (BMI only, $r_s$=0.46, p=0.046) (S15 Table). As expected, both weight and BMI reduction correlated with the increase in urinary ketones ($r_s$=-0.56, p=0.012 and $r_s$ =-0.51, p=0.025, respectively) and also with the reduction in urinary mannitol ($r_s$=0.540, p=0.014 and $r_s$=0.498, p=0.026, respectively) (S15 Table).

FFQ showed that the reduction in abdominal circumference was significantly associated with a decrease in the consumption of cereals and derivatives and processed meats ($r_s$=0.51, p=0.022 and $r_s$=0.52, p=0.020, respectively). Weight and BMI reduction correlated with blood pressure reduction: $r_s$=0.52, p=0.020 and $r_s$=0.53, p=0.017 with SBP and $r_s$=0.48, p=0.031 weight with DBP), but no significant associations were found with cardiovascular indices modification (S17 Table).

## Association of variables at W0 with modifications during the study (identifying predictors of better response to VLCKD)

The results of the associations between variables at W0 and variable improvement during the study are provided in Fig 3 and in detail in S19–S27 Tables.

Individuals with greater weight at W0 lost more weight during the study (S25 Table). Also, participants with higher disease activity at W0, as measured by DAPSA and DAS28-CRP, showed greater improvements in all anthropometric measures. Subjects with elevated IL-6 at W0, who also had higher disease activity at W0, exhibited greater improvements in weight and abdominal circumference (S19 Table); on the contrary, individuals with higher TNF-α at

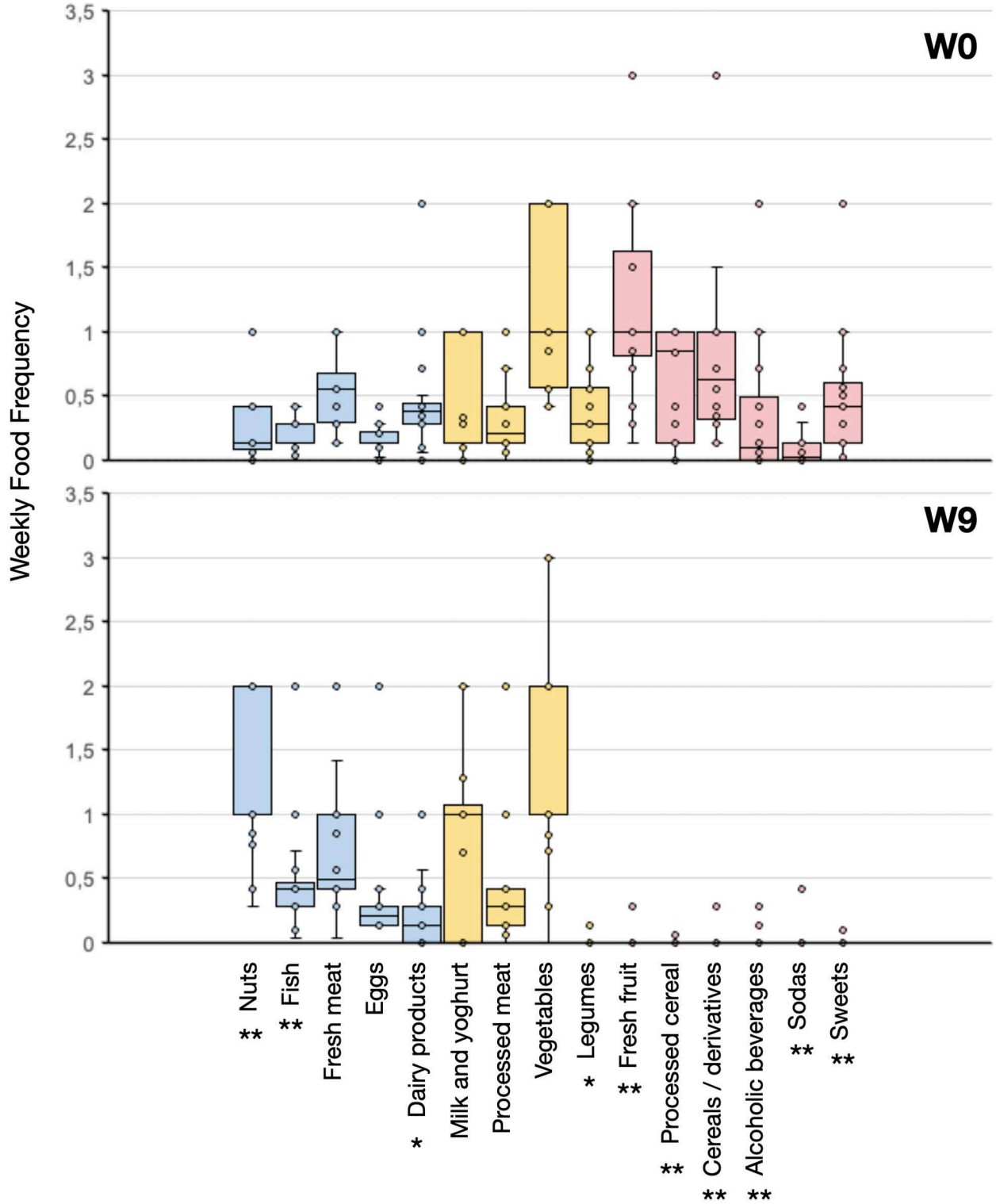

**Fig 1. Food frequency modification during the study according to the food frequency questionnaire.** Significance refers to the Wilcoxon test, * **p** < 0.05, **p < 0.01. W0 week 0; W9 week 9. Although SBP and DBP were not significantly reduced, improvement in cardiovascular risk was significant: -0.2 (-0.7;0.1) units in SCORE2 10-year risk percentage.

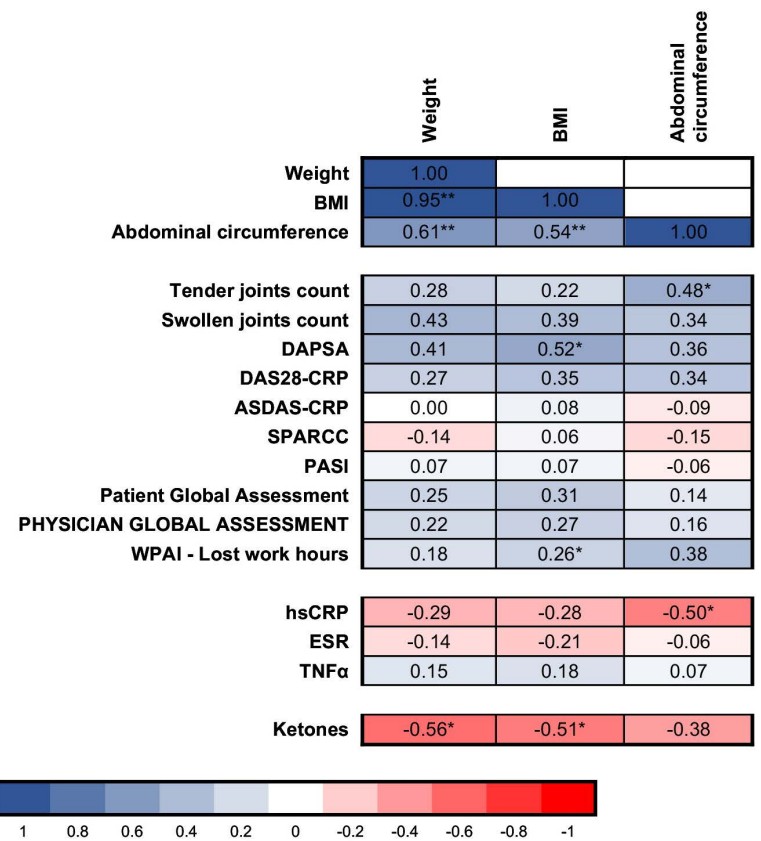

**Fig 2. Correlation matrix between modification of anthropometric measurements and modification of other variables during the study.** Values displayed refer to Spearman's correlation coefficient r$_s$; negative correlations are in red, positive correlations are in blue. * **p** < 0.05, **p < 0.01, BMI, body mass index, DAPSA, disease activity in psoriatic arthritis; DAS28-CRP, disease activity score 28 with C reactive protein; ASDAS-CRP, Ankylosing Spondylitis Disease Activity Score – CRP; SPARCC, Spondyloarthritis Research Consortium of Canada; PASI, Psoriasis Area Severity Index; Physician Global Assessment; WPAI, Work Productivity and Activity Impairment questionnaire; hsCRP, High sensitivity CRP, ESR, Erythrocyte Sedimentation Rate; IL-1α; TNFα, Tumor Necrosis Factor alpha.

W0, who also presented lower disease activity at W0, showed a significantly smaller reduction in weight and BMI during the study (S25 Table). Subjects with higher activity and higher ESR and hsCRP at W0, experienced significantly greater improvements in almost all disease activity measures (S25 Table).

## Discussion

Our preliminary study showed that a 9-week VLCKD, a strict low-calorie, and low-carbohydrate nutritional regimen, is feasible for overweight and obese patients with PsA. Furthermore, the VLCKD was very effective in reducing weight, with >10% weight loss observed in our patients, and it improved most domains of PsA activity. A strict adherence to the diet improved insulin resistance, lipid profiles, and blood pressure parameters, thus reducing cardiovascular risk. Considering that disease activity in all patients in the study was low at baseline, no significant modifications of inflammatory parameters were observed.

All patients adhered to the carbohydrate restriction protocol, as indicated by the increased levels of urinary ketones, demonstrating the effectiveness of the nutritional approach which included evaluations every three weeks. Additionally, ketosis effectively

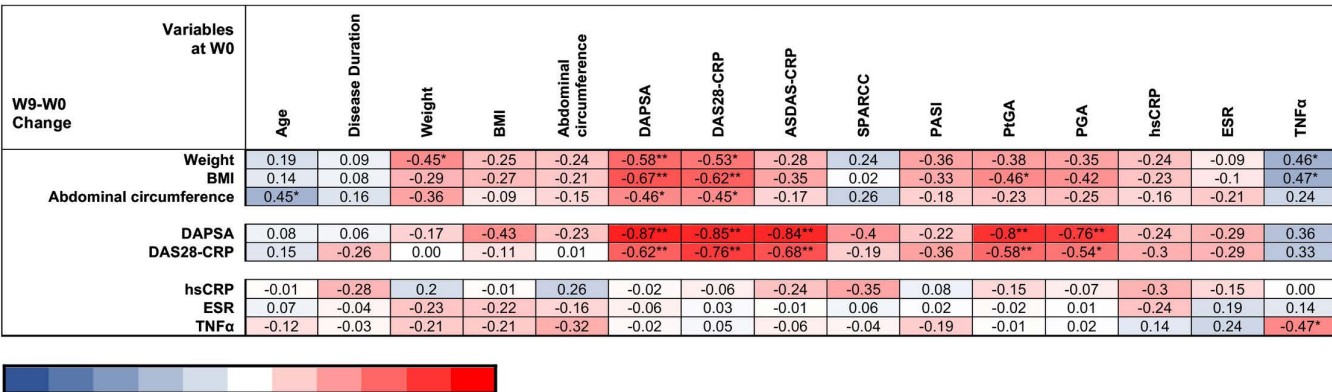

| W9-W0 Change | Age | Disease Duration | Weight | BMI | Abdominal circumference | DAPSA | DAS28-CRP | ASDAS-CRP | SPARCC | PASI | PtGA | PGA | hsCRP | ESR | TNFα |
|---|---|---|---|---|---|---|---|---|---|---|---|---|---|---|---|
| Weight | 0.19 | 0.09 | -0.45* | -0.25 | -0.24 | -0.58** | -0.53* | -0.28 | 0.24 | -0.36 | -0.38 | -0.35 | -0.24 | -0.09 | 0.46* |
| BMI | 0.14 | 0.08 | -0.29 | -0.27 | -0.21 | -0.67** | -0.62** | -0.35 | 0.02 | -0.33 | -0.46* | -0.42 | -0.23 | -0.1 | 0.47* |
| Abdominal circumference | 0.45* | 0.16 | -0.36 | -0.09 | -0.15 | -0.46* | -0.45* | -0.17 | 0.26 | -0.18 | -0.23 | -0.25 | -0.16 | -0.21 | 0.24 |
| | | | | | | | | | | | | | | | |
| DAPSA | 0.08 | 0.06 | -0.17 | -0.43 | -0.23 | -0.87** | -0.85** | -0.84** | -0.4 | -0.22 | -0.8** | -0.76** | -0.24 | -0.29 | 0.36 |
| DAS28-CRP | 0.15 | -0.26 | 0.00 | -0.11 | 0.01 | -0.62** | -0.76** | -0.68** | -0.19 | -0.36 | -0.58** | -0.54* | -0.3 | -0.29 | 0.33 |
| | | | | | | | | | | | | | | | |
| hsCRP | -0.01 | -0.28 | 0.2 | -0.01 | 0.26 | -0.02 | -0.06 | -0.24 | -0.35 | 0.08 | -0.15 | -0.07 | -0.3 | -0.15 | 0.00 |
| ESR | 0.07 | -0.04 | -0.23 | -0.22 | -0.16 | -0.06 | 0.03 | -0.01 | 0.06 | 0.02 | -0.02 | 0.01 | -0.24 | 0.19 | 0.14 |
| TNFα | -0.12 | -0.03 | -0.21 | -0.21 | -0.32 | -0.02 | 0.05 | -0.06 | -0.04 | -0.19 | -0.01 | 0.02 | 0.14 | 0.24 | -0.47* |

**Fig 3. Correlation matrix between variables at W0 and the modification of main variables during the study.** Values displayed refer to Spearman's correlation coefficient $r_s$ negative correlations are in red, positive correlations are in blue. * $p < 0.05$, ** $p < 0.01$, BMI, body mass index, DAPSA, disease activity index in psoriatic arthritis; DAS28-CRP, disease activity score 28 with C reactive protein; ASDAS-CRP, Ankylosing Spondylitis Disease Activity Score – CRP; SPARCC, Spondyloarthritis Research Consortium of Canada; PASI, Psoriasis Area Severity Index; WPAI, Work Productivity and Activity Impairment questionnaire; hsCRP, High-sensitivity CRP, ESR, Erythrocyte Sedimentation Rate; IL-1α; TNFα, Tumor Necrosis Factor alpha.

suppresses hunger and enhances satiety, which may therefore constitute an advantage over the hypocaloric Mediterranean diet and improve adherence [38]. Patients with higher baseline weights showed greater motivation and achieved more significant weight loss suggesting that individuals with obesity, rather than overweight, may be more suitable candidates for VLCKD.

The study revealed improvements across nearly all indices of PsA activity, including peripheral arthritis, axial involvement, and psoriasis. DAPSA score decreased significantly, especially in those who exhibited more substantial BMI reductions. These changes did not stem from reduced biohumoral markers, but from the significant improvement in tender joint count, pain scores, and patient-reported outcomes. Improved functionality was also evidenced by fewer lost working hours, which also correlated with BMI reduction. The primary aim of a VLCKD is to induce significant weight loss, which reduces mechanical stress on the joints and, in turn, decreases pain and inflammation [39]. Consequently, disease activity is diminished even in the absence of a reduction in inflammatory mediators, as observed in our study, where patients exhibited normal levels of inflammatory biomarkers. This finding explains the lack of association between disease activity modification and changes in biohumoral markers. Nonetheless, although VLCKD lacks specific food component intake indications, it may exert putative anti-inflammatory and immunomodulatory effects on mediators not analyzed in this study. Notably, VLCKD and KD have also been observed to improve skin disease in psoriatic patients [24,25].

Despite evidence of anti-inflammatory biological pathways associated with VLCKD and previous reports indicating reduced levels of TNFα and IL-6 following the diet [19,28,29,30,31,32,40,41], only a few changes in inflammatory biomarkers were observed in this study. Likewise, the sole previous study in the literature that examined the implementation of KD in PsA patients, demonstrated that KD specifically reduced levels of IL-6, IL-17 and IL-23, but not other inflammatory biomarkers [34]. The lack of significant changes in serological inflammation should be interpreted in the context of the baseline characteristics of our study population, which consisted of subjects with well-controlled disease. Most were receiving bDMARDs, and baseline inflammatory cytokine levels were below the measurable limits for many subjects.

High baseline TNF-alpha levels were predictive of less significant weight loss, while high baseline IL-6 levels appeared predictive of more significant weight loss. A possible explanation for this finding is that the study may have lacked sufficient statistical power to detect significant changes in specific biomarkers, given the heterogeneous nature of the population. Indeed, different disease pattern may be present at W0: those with high TNF-α levels, associated with low disease activity indices and lower BMI, and those with high IL-6 levels, correlated with high disease activity indices, particularly in relation to joint involvement and higher BMI. Patients with elevated TNF-α levels and lower BMI at W0 may exhibit a different pattern of psoriatic disease, characterized by milder joint involvement and potentially a reduced benefit from VLCKD.

Changes in the intestinal transcellular pathway may affect the intestinal permeability, measured by the lactulose/mannitol ratio which appeared to worsen — i.e., increased urinary mannitol — though it remained within the reference range. This phenomenon has been previously described in obese patients undergoing VLCKD but requires further investigation [42].

The lipid and metabolic profiles improved significantly in all patients, as expected with KD [43]. Notably, we also noted a significant reduction in insulin resistance — over 50% vs. baseline — as assessed by the HOMA-IR index, in line with previous reports in the literature and reinforcing the beneficial metabolic effects of this dietary approach [44].

A major strength of our study is that we assessed cardiovascular risk over a short time period after implementing VLCKD, whereas such effects were typically expected to manifest over a longer period [45]. Cardiovascular risk indices showed a slight albeit significant improvement overall. This is particularly noteworthy as these indices do not incorporate anthropometric measurements; the improved lipid profiles and blood pressure correlated with weight and BMI reduction.

Despite a drastic reduction in carbohydrates (including legumes and fresh fruit), the VLCKD did not reduce the adherence to the Mediterranean diet because of an increased intake of fish and nuts. This is important given the known benefits of the Mediterranean diet constituents for rheumatic patients and those at higher cardiovascular risk.

We would be remiss if we did not mention some of the limitations of our study. The small sample size and short observation period may have limited the statistical power and generalizability of our findings. Furthermore, the absence of a control arm prevents a comprehensive analysis of confounders which makes it difficult to determine whether the observed effects are attributable to weight loss itself or specifically to the VLCKD protocol compared to other dietary interventions. Nonetheless, given the highly restrictive nature of this diet—intended for short-term use under clinical supervision— and the high likelihood of dropouts, a proof-of-concept study was deemed appropriate to evaluate the potential benefits of this dietary approach. These findings provide a foundation for future investigations. We did not measure additional inflammatory cytokines involved in PsA, such as IL-17, which could have helped detect subtle improvements in the serological profile and inflammatory state of patients. A larger-scale study will also allow for a more thorough analysis of confounders, particularly in relation to DMARD treatment and patients with high disease activity.

Our study demonstrated that VLCKD was effective in improving anthropometric measures, PsA activity, and metabolic and cardiovascular parameters in overweight and obese patients, with the degree of improvement correlating with BMI reduction. Further trials are necessary to investigate the combination of the VLCKD with nutritional regimens that could sustain these short-term benefits over time, gradually transitioning the patient back to a Mediterranean dietary pattern, which should remain the reference model for chronic inflammatory arthritis. Additionally, the study underscores the importance of diet as an adjunct to pharmacological treatments, with the potential to reduce the cardiovascular and metabolic burden in patients with PsA.

## Supporting information

**S1 Table. Full list of variables included in the study.**
(PDF)

**S2 Table. Characteristics of the patients at W0.**
(PDF)

**S3 Table. Association between continuous and categorical variables at W0.**
(PDF)

**S4 Table. Association between continuous variables at W0.**
(PDF)

**S5 Table: Association between categorical variables at W0.**
(PDF)

**S6 Table. Modification of anthropometric measurements during the study.**
(PDF)

**S7 Table. Modification of clinical variables during the study.**
(PDF)

**S8 Table. Modification of inflammatory biomarkers during the study.**
(PDF)

**S9 Table. Modification of laboratory variables during the study.**
(PDF)

**S10 Table. Modification of nutritional questionnaires during the study.**
(PDF)

**S11 Table. Modification of cardiovascular parameters during the study.**
(PDF)

**S12 Table. Correlation between the modification of anthropometric measurements during the study.**
(PDF)

**S13 Table. Correlation between the modification of anthropometric measurements and the modification of clinical variables during the study.**
(PDF)

**S14 Table. Correlation between the modification of anthropometric measurements and the modification of inflammatory biomarkers during the study.**
(PDF)

**S15 Table. Correlation between the modification of anthropometric measurements and the modification of laboratory variables during the study.**
(PDF)

**S16 Table. Correlation between the modification of anthropometric measurements and the modification of food frequencies during the study.**
(PDF)

**S17 Table. Correlation between the modification of anthropometric measurements and the modification of cardiovascular parameters during the study.**
(PDF)

**S18 Table. Association between the modification of anthropometric measurements and the modification of categorical variables during the study.**
(PDF)

**S19 Table. Analysis of the association between categorical variables at W0 and the modification of continuous anthropometric measurements during the study.**
(PDF)

**S20 Table. Analysis of the association between categorical variables at W0 and the modification of continuous clinical variables during the study.**
(PDF)

**S21 Table. Analysis of the association between categorical variables at W0 and the modification of inflammatory biomarkers during the study.**
(PDF)

**S22 Table. Analysis of the association between categorical variables at W0 and the modification of continuous laboratory variables during the study.**
(PDF)

**S23 Table. Analysis of the association between categorical variables at W0 and the modification of nutritional questionnaires during the study.**
(PDF)

**S24 Table. Analysis of the association between categorical variables at W0 and the modification of cardiovascular parameters during the study.**
(PDF)

**S25 Table. Association between continuous variables at W0 and the modification of continuous variables during the study.**
(PDF)

**S26 Table. Association between categorical variables at W0 and the modification of categorical variables (clinical, inflammatory biomarkers, and cardiovascular parameters).**
(PDF)

**S27 Table. Association between continuous variables at W0 and the modification of categorical variables during the study.**
(PDF)

**S1 Figure. Simple Boxplots of permeability tests during the study.**
(PDF)

## Acknowledgements

The authors would like to thank nurses Nicoletta Pedron and Federica Delon for their invaluable contributions that led to the amelioration of the study.

## Author contributions

**Conceptualization:** Roberta Ramonda, Giovanni Striani, Daniela Basso, Mariagrazia Lorenzin, Marta Favero, Filippo Brocadello, Andrea Doria.

**Data curation:** Mariagrazia Lorenzin, Laura Scagnellato, Ada Aita, Filippo Brocadello.

**Formal analysis:** Francesca Ometto, Giovanni Striani, Filippo Evangelista, Ada Aita, Marta Favero.

**Investigation:** Giovanni Striani, Giacomo Cozzi, Daniela Basso, Filippo Evangelista, Mariagrazia Lorenzin, Laura Scagnellato, Filippo Brocadello.

**Methodology:** Roberta Ramonda, Francesca Ometto, Marta Favero.

**Project administration:** Giovanni Striani, Giacomo Cozzi.

**Resources:** Giovanni Striani, Giacomo Cozzi, Daniela Basso, Laura Scagnellato, Ada Aita.

**Supervision:** Roberta Ramonda, Francesca Ometto, Marta Favero, Andrea Doria.

**Validation:** Roberta Ramonda, Andrea Doria.

**Visualization:** Marta Favero.

**Writing – original draft:** Francesca Ometto, Giovanni Striani.

**Writing – review & editing:** Roberta Ramonda, Francesca Ometto, Andrea Doria.

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
