## [Decision Letter · Decision Letter 0]

21 Jan 2025

PONE-D-24-38756Ketogenic diet improves disease activity and cardiovascular risk in psoriatic arthritis: a proof of concept studyPLOS ONE

Dear Dr. Ramonda,

Thank you for submitting your manuscript to PLOS ONE. After careful consideration, we feel that it has merit but does not fully meet PLOS ONE’s publication criteria as it currently stands. Therefore, we invite you to submit a revised version of the manuscript that addresses the points raised during the review process.

We look forward to receiving your revised manuscript.

Kind regards,

Luca Navarini

Academic Editor

PLOS ONE

**Journal Requirements:**

Reviewers' comments:

Reviewer's Responses to Questions

**Comments to the Author**

1. Is the manuscript technically sound, and do the data support the conclusions?

Reviewer #1: Yes

Reviewer #2: Partly

2. Has the statistical analysis been performed appropriately and rigorously? 

Reviewer #1: Yes

Reviewer #2: N/A

3. Have the authors made all data underlying the findings in their manuscript fully available?

Reviewer #1: Yes

Reviewer #2: Yes

4. Is the manuscript presented in an intelligible fashion and written in standard English?

Reviewer #1: Yes

Reviewer #2: Yes

5. Review Comments to the Author

**Reviewer #1:**  The manuscript addresses an important aspect of PsA management by focusing on a very low-calorie ketogenic diet, that has been suggested to have anti-inflammatory effects. The article is well-written and organized, although correction for minor typos is encouraged. I have a few comments on the paper.

1) The authors state that the VLCKD is associated with an improvement of disease activity, which correlates with BMI reduction. However, this improvement is not associated with changes in IL-6 and TNFα levels. How do the authors explain this result?

2) Authors say that individuals with higher TNFα at W0 showed greater weight and BMI reduction if they had low disease activity. Why according to the authors are TNFα levels inversely correlated with disease activity?

3) Although there are only two patients with diabetes, did the authors find any differences in the effectiveness of the VLCKD compared to the remaining study cohort?

4) The authors affirm that patients with higher disease activity at W0 according to DAPSA and DAS28-CRP showed greater improvements in blood glucose levels and HOMA-IR. What correlation do the authors think there is between DAPSA/DAS28 and blood glucose concentration/HOMA-IR?

It will be interesting in the future to include patients with high disease activity at baseline, as well as to evaluate variations in IL17 in patients undergoing a VLCKD.

**Reviewer #2:**  This manuscript presents data from PSA patients undergoing a very low calories ketogenic diet (VLCKD). Changes in anthropometric measures, BMI, and lipid profile are correlated with changes in disease activity. Major issues to be addressed include the following:

1. The main limitation is the lack of a control arm, preventing any consideration about the efficacy of the VLKD itself. Only correlations between weight loss/ diet adherence and disease activity parameters can be examined, but it is impossible to understand whether observed changes are due to the specific type of diet, or to weight loss in general. In fact, similar results have been obtained with different types of diet, in studies with similar limitations (Klingberg E, et al. Arthritis Res Ther. 2019;21(1):17; Klingberg E, et al. Arthritis Res Ther. 2020;22(1):254)

2. Sample size is also quite small (n=20) without specific considerations as to why this was considered to be adequate.

3. The heterogeneity of the included population, e.g. in terms of background treatment, poses a challenge in controlling for potential confounders. In fact, as the authors note, inflammatory biomarkers were significantly lower in the bDMARDs group vs csDMARDs only. This observation would suggest it would be more appropriate to include patients with a more similar background treatment, or at least to control for therapy, which seems to be an important confounder, in the analyses. Nonetheless, this is a very difficult task with a sample size of 20 patients.

4. A high number of analyses have been carried out, some without a real objective (e.g in figure 2 correlation between changes in BMI and anthropometric measures, or in figure 3 correlations between various disease activity indexes, both quite expected and without an added value to existing knowledge), increasing significantly type I error. I would suggest removing these kind of analysis and focusing on the correlations between changes in BMI/anthropometric measures/ketones on one side, and disease activity or inflammatory indexes on the other side.

In summary the topic of the manuscript is potentially relevant, however, the limited sample size, the lack of a control arm and control for confounders, significantly limit the impact of results.

6. PLOS authors have the option to publish the peer review history of their article (what does this mean? ). If published, this will include your full peer review and any attached files.

**Do you want your identity to be public for this peer review?** For information about this choice, including consent withdrawal, please see our Privacy Policy .

Reviewer #1: No

Reviewer #2: No

---

## [Author Response · Author response to Decision Letter 1]

30 Jan 2025

Reviewer #1

The manuscript addresses an important aspect of PsA management by focusing on a very low-calorie ketogenic diet, that has been suggested to have anti-inflammatory effects. The article is well-written and organized, although correction for minor typos is encouraged. I have a few comments on the paper.

1) The authors state that the VLCKD is associated with an improvement of disease activity, which correlates with BMI reduction. However, this improvement is not associated with changes in IL-6 and TNFα levels. How do the authors explain this result?

> We thank the reviewer for highlighting this important issue. The intricate relationship between BMI modification, disease activity, and inflammatory indices is indeed complex and deserves further explanation. We have elaborated on our hypotheses and revised a relevant paragraph in the discussion section to provide greater clarity as follows:

“The primary aim of a VLCKD is to induce significant weight loss, which reduces mechanical stress on the joints and, in turn, decreases pain and inflammation. Consequently, disease activity is diminished even in the absence of a reduction in inflammatory mediators, as observed in our study, where patients exhibited normal levels of inflammatory biomarkers. This finding explains the lack of association between disease activity modification and changes in biohumoral markers. Nonetheless, although VLCKD lacks specific food component intake indications, it may exert putative anti-inflammatory and immunomodulatory effects on mediators not analyzed in this study. Notably, VLCKD and KD have also been observed to improve skin disease in psoriatic patients.”

2) Authors say that individuals with higher TNFα at W0 showed greater weight and BMI reduction if they had low disease activity. Why according to the authors are TNFα levels inversely correlated with disease activity?

> We thank the reviewer for highlighting this issue and encouraging us to clarify this observation and improve our text. In response, we have included a new paragraph in the discussion section, adding further clarification as follows:

“High baseline TNF-alpha levels were predictive of less significant weight loss, while high baseline IL-6 levels appeared predictive of more significant weight loss. A possible explanation for this finding is that the study may have lacked sufficient statistical power to detect significant changes in specific biomarkers, given the heterogeneous nature of the population. Indeed, different disease pattern may be present at W0: those with high TNF-α levels, associated with low disease activity indices and lower BMI, and those with high IL-6 levels, correlated with high disease activity indices, particularly in relation to joint involvement and higher BMI. Patients with elevated TNF-α levels and lower BMI at W0 may exhibit a different pattern of psoriatic disease, characterized by milder joint involvement and potentially a reduced benefit from VLCKD.”

3. Although there are only two patients with diabetes, did the authors find any differences in the effectiveness of the VLCKD compared to the remaining study cohort?

> We thank the reviewer for drawing attention to this observation. Two patients with diabetes were included in the study; however, their data were excluded due to a lack of significant findings and the redundancy of the information already provided by insulin levels and HOMA-IR. For completeness, we have included the data (with no significant association observed) between diabetes and changes in anthropometric measures in Supplementary Table S21.

4. The authors affirm that patients with higher disease activity at W0 according to DAPSA and DAS28-CRP showed greater improvements in blood glucose levels and HOMA-IR. What correlation do the authors think there is between DAPSA/DAS28 and blood glucose concentration/HOMA-IR?

>We appreciate the reviewer for highlighting this issue. Higher activity indices at W0 were associated with higher weight, and these patients exhibited greater improvements in anthropometric measures compared to those with lower activity indices. Anthropometric improvement, in turn, was linked to an amelioration of insulin resistance, which explains the observed association between high activity indices at W0 and improved insulin resistance. Nonetheless, this effect is likely attributable to weight loss. To avoid confounding and redundancy, as suggested by Reviewer #2, we have focused on the most relevant findings and removed this information from the last paragraph of the results section. Data, however, remain available in the supplementary information.

Participants with higher disease activity at W0 according to DAPSA and DAS28-CRP showed greater improvements in blood glucose levels and HOMA-IR. Participants with higher BMI and weight at W0 showed more significant reductions in WBC, lymphocytes, blood glucose and HOMA-IR.

5. It will be interesting in the future to include patients with high disease activity at baseline, as well as to evaluate variations in IL17 in patients undergoing a VLCKD.

> We thank the reviewer for highlighting this issue. These limitations were already addressed in the discussion and have now been further emphasized as follows:

“We did not measure additional inflammatory cytokines involved in PsA, such as IL-17, which could have helped detect subtle improvements in the serological profile and inflammatory state of patients.”

Reviewer #2

This manuscript presents data from PSA patients undergoing a very low calories ketogenic diet (VLCKD). Changes in anthropometric measures, BMI, and lipid profile are correlated with changes in disease activity. Major issues to be addressed include the following:

1. The main limitation is the lack of a control arm, preventing any consideration about the efficacy of the VLKD itself. Only correlations between weight loss/ diet adherence and disease activity parameters can be examined, but it is impossible to understand whether observed changes are due to the specific type of diet, or to weight loss in general. In fact, similar results have been obtained with different types of diet, in studies with similar limitations (Klingberg E, et al. Arthritis Res Ther. 2019;21(1):17; Klingberg E, et al. Arthritis Res Ther. 2020;22(1):254)

>We thank the reviewer for highlighting this important limitation. We have added this point to the limitation paragraph in the discussion and rephrased the entire paragraph as follows:

"The small sample size and short observation period may have limited the statistical power and generalizability of our findings. Furthermore, the absence of a control arm prevents a comprehensive analysis of confounders which makes it difficult to determine whether the observed effects are attributable to weight loss itself or specifically to the VLCKD protocol compared to other dietary interventions. Nonetheless, given the highly restrictive nature of this diet—intended for short-term use under clinical supervision— and the high likelihood of dropouts, a proof-of-concept study was deemed appropriate to evaluate the potential benefits of this dietary approach. These findings provide a foundation for future investigations. We did not measure additional inflammatory cytokines involved in PsA, such as IL-17, which could have helped detect subtle improvements in the serological profile and inflammatory state of patients. A larger-scale study will also allow for a more thorough analysis of confounders, particularly in relation to DMARD treatment and patients with high disease activity.”

Also references were added in the text in the discussion section:

Reference number 40:

“The primary aim of a VLCKD is to induce significant weight loss, thereby reducing mechanical stress on the joints, which leads to decreased pain and inflammation40.”

40. Klingberg E, Bilberg A, Björkman S, Hedberg M, Jacobsson L, Forsblad-d'Elia H, Carlsten H, Eliasson B, Larsson I. Weight loss improves disease activity in patients with psoriatic arthritis and obesity: an interventional study. Arthritis Res Ther. 2019;21(1):17.

Reference number 46:

“A major strength of our study is that we assessed cardiovascular risk over a short time period after implementing VLCKD, such effects were typically expected to manifest over a longer period46”

46. Klingberg E, Björkman S, Eliasson B, Larsson I, Bilberg A. Weight loss is associated with sustained improvement of disease activity and cardiovascular risk factors in patients with psoriatic arthritis and obesity: a prospective intervention study with two years of follow-up. Arthritis Res Ther. 2020;22(1):254.

2. Sample size is also quite small (n=20) without specific considerations as to why this was considered to be adequate.

>We thank the reviewer for pointing out this issue. As discussed above, the study is a proof-of-concept study which is adequate for a very restrictive diet. The issue has been addressed in the limitations paragraph, specifically as follows:

“The small sample size and short observation period may have limited the statistical power and generalizability of our findings. […] Nonetheless, given the highly restrictive nature of this diet—intended for short-term use under clinical supervision— and the high likelihood of dropouts, a proof-of-concept study was deemed appropriate to evaluate the potential benefits of this dietary approach.”

3. The heterogeneity of the included population, e.g. in terms of background treatment, poses a challenge in controlling for potential confounders. In fact, as the authors note, inflammatory biomarkers were significantly lower in the bDMARDs group vs csDMARDs only. This observation would suggest it would be more appropriate to include patients with a more similar background treatment, or at least to control for therapy, which seems to be an important confounder, in the analyses. Nonetheless, this is a very difficult task with a sample size of 20 patients.

>We thank the reviewer for pointing out this issue. We agree with their observation and have emphasized this point in the limitations section:

“A larger-scale study will also allow for a more thorough analysis of confounders, particularly in relation to DMARD treatment and patients with high disease activity.”

4. A high number of analyses have been carried out, some without a real objective (e.g in figure 2 correlation between changes in BMI and anthropometric measures, or in figure 3 correlations between various disease activity indexes, both quite expected and without an added value to existing knowledge), increasing significantly type I error. I would suggest removing these kind of analysis and focusing on the correlations between changes in BMI/anthropometric measures/ketones on one side, and disease activity or inflammatory indexes on the other side.

> We thank the reviewer for contributing to the improvement of the clarity of our findings. We agree with their observation and have summarized findings that were less relevant and could limit the comprehension of the main results. Information regarding activity indices, inflammatory biomarkers, and anthropometric measures has been highlighted, while redundant information and details about laboratory tests and questionnaires have been summarized as follows:

In section “Association between variables at W0” the following sentences were removed or rephrased:

“Notably, higher TNFα levels also associated with higher f-CPT levels, and both of them displayed significantly lower levels in subjects treated with bDMARDs. As, expected, fasting insulinemia, glucose and HOMA-IR positively correlated with weight, BMI and abdominal circumference, being higher in individuals with increased measurements : HOMA-IR rs=0.50, p=0.042; rs=0.54, p=0.001 and rs=0.66, p=0.49, p=0.028, respectively” (Supplementary Table S4b). At W0, anthropometric measurements showed no significant associations with the lipid profile or the cardiovascular indices (Supplementary Tables S4b and S4c).”

In section “Changes between W0 and W9 (identification of significant modifications following VLCKD)” the following sentence was removed:

“Notably, WBC and platelet count significantly decreased (Supplementary Table S9).”

In section“Association between changes in anthropometric measurements and changes in other variables during the study (assessing the impact of weight, BMI and abdominal circumference modification on PsA disease activity)” the following sentences were removed or additional information was included:

“No significant correlation between changes in anthropometric measures and inflammation indices was observed, except for a modest and inverse association between the change in abdominal circumference and the change in hsCRP (rs=0.50, p=0.046) (Supplementary Tables S15 and S19)”

“Greater weight reduction was also associated with a decrease in neutrophils, and BMI with WBC (Supplementary Table S16)”

“Weight and BMI reduction correlated with blood pressure reduction (rs=0.52, p=0.020 and rs=0.53, p=0.017, respectively with SBP and rs=0.48, p=0.031 weight with DBP), but no significant associations were found with cardiovascular indices modification (Supplementary Table S18)

In section “Association of variables at W0 with modifications during the study (identifying predictors of better response to VLCKD)” the following paragraphs were removed or rephrased:

“Furthermore, those with greater weight and abdominal circumference at W0 also experienced a greater reduction in TJC and weight decrease correlated with the reduction of WPAI lost work hours. Subjects with higher ESR and hsCRP at W0, who also presented higher DAS28-CRP, showed significantly greater improvements in TJC, SJC, and axial disease measures (ASDAS-CRP) (Supplementary Table S27a).” ).

“Subjects with higher activity and higher ESR and hsCRP at W0, experienced significantly greater improvements in almost all disease activity measures (Supplementary Table S27a and S27b).”

Participants with higher disease activity at W0 according to DAPSA and DAS28-CRP showed greater improvements in blood glucose levels and HOMA-IR. Participants with higher BMI and weight at W0 showed more significant reductions in WBC, lymphocytes, blood glucose and HOMA-IR. Subjects with elevated IL-6 at W0 more frequently showed improvement in HDL cholesterol; subjects with elevated TNF-α levels more frequently presented a reduction in TC and LDL cholesterol and those with elevated f-CPT had a more frequent reduction in LDL cholesterol (Supplementary Table S23). Subjects with higher cardiovascular risk at W0, according to SCORE2, improved more in the same index. Participants with higher BMI and weight at W0 showed more significant reductions in SBP and DBP.”

In the discussion section the following sentence was removed:

Notably, IL-6 levels, as well as platelet and white blood cell counts were decreased as observed previously

Figures 2 and 3 have also been summarized as suggested. Cardiovascular indices and laboratory tests were removed. Information regarding anthropometric measures, disease activity, and inflammatory biomarkers has been retained and organized to improve clarity.

6. In summary the topic of the manuscript is potentially relevant, however, the limited sample size, the lack of a control arm and control for confounders, significantly limit the impact of results.

>We further thank the reviewer for their interest in our paper. We have followed the reviewer’s suggestion and modified substantially the result section by removing less relevant information and highlighting the major findings. Furthermore, we are aware that the study population is small, nonetheless being the diet very restrictive a proof-of-concept study was deemed necessary in PSA patients before a larger-scale study was conducted. We also discussed the limitations of our study more comprehensively in the discussion section, as outlined above (points 1, 2 and 3).

---

## [Decision Letter · Decision Letter 1]

2 Mar 2025

Ketogenic diet improves disease activity and cardiovascular risk in psoriatic arthritis: a proof of concept study

PONE-D-24-38756R1

Dear Dr. Ramonda,

We’re pleased to inform you that your manuscript has been judged scientifically suitable for publication and will be formally accepted for publication once it meets all outstanding technical requirements.

Kind regards,

Luca Navarini

Academic Editor

PLOS ONE

Additional Editor Comments (optional):

Reviewers' comments:

Reviewer's Responses to Questions

**Comments to the Author**

1. If the authors have adequately addressed your comments raised in a previous round of review and you feel that this manuscript is now acceptable for publication, you may indicate that here to bypass the “Comments to the Author” section, enter your conflict of interest statement in the “Confidential to Editor” section, and submit your "Accept" recommendation.

Reviewer #1: (No Response)

2. Is the manuscript technically sound, and do the data support the conclusions?

Reviewer #1: Yes

3. Has the statistical analysis been performed appropriately and rigorously? 

Reviewer #1: Yes

4. Have the authors made all data underlying the findings in their manuscript fully available?

Reviewer #1: Yes

5. Is the manuscript presented in an intelligible fashion and written in standard English?

Reviewer #1: Yes

6. Review Comments to the Author

Reviewer #1: thanks to the authors for the modifications and clarifications which have allowed an improvement of the paper.

7. PLOS authors have the option to publish the peer review history of their article (what does this mean? ). If published, this will include your full peer review and any attached files.

**Do you want your identity to be public for this peer review?** For information about this choice, including consent withdrawal, please see our Privacy Policy .

Reviewer #1: No

---

## [Editor Report · Acceptance letter]

PONE-D-24-38756R1

PLOS ONE

Dear Dr. Ramonda,

I'm pleased to inform you that your manuscript has been deemed suitable for publication in PLOS ONE. Congratulations! Your manuscript is now being handed over to our production team.

Kind regards,

on behalf of

Dr. Luca Navarini

Academic Editor

PLOS ONE